# The Healthfulness of the US Packaged Food and Beverage Supply: A Cross-Sectional Study

**DOI:** 10.3390/nu11081704

**Published:** 2019-07-24

**Authors:** Abigail S. Baldridge, Mark D. Huffman, Fraser Taylor, Dagan Xavier, Brooke Bright, Linda V. Van Horn, Bruce Neal, Elizabeth Dunford

**Affiliations:** 1Department of Preventive Medicine, Northwestern University Feinberg School of Medicine, Chicago, IL 60611, USA; 2The George Institute for Global Health, University of New South Wales, Sydney, NSW 2042, Australia; 3Label Insight, Chicago, IL 60661, USA

**Keywords:** processed food, food supply, Health Star Rating, nutrient profiling, NOVA classification

## Abstract

The US food supply is dominated by highly-processed packaged food and beverage products that are high in energy, saturated fat, sugar, and salt. We report results of a cross-sectional assessment of the 2018 US packaged food and beverage supply by nutritional composition and indicators of healthfulness and level of processing. Data were obtained through Label Insight’s Open Data database, which represents >80% of all food and beverage products sold in the US over the past three years. Healthfulness and the level of processing, measured by the Health Star Rating (HSR) system and the NOVA classification framework, respectively, were compared across product categories and leading manufacturers. Among 230,156 food and beverage products, the mean HSR was 2.7 (standard deviation (SD) 1.4) from a possible maximum rating of 5.0, and 71% of products were classified as ultra-processed. Healthfulness and level of processing varied substantially by category (range: HSR 1.1–3.9; 0–100% ultra-processed) and manufacturer (range: HSR 0.9–4.6; 26–100% ultra-processed). The US packaged food and beverage supply is large, heterogeneous, highly processed, and generally unhealthy. The wide variability in healthfulness and level of processing demonstrates that opportunities exist, through reformulation or replacement, for large-scale improvements to the healthfulness of the US packaged food and beverage supply.

## 1. Introduction

The central role of the food supply in the development of chronic disease is well recognized [1,2]. The US food supply is dominated by packaged food and beverage products, with around 80% of total calories consumed coming from store-bought foods and beverages [3,4]. As such, the US population is being exposed to products that are high in energy, saturated fat, sugar, and salt [1]. Even small changes in levels of these nutrients of concern in the food supply have the potential to produce large health gains at relatively low cost, and these changes are being promoted by public health experts as priority actions to address the growing global chronic disease crisis [5]. Food and beverage manufacturers play an important role in not only creating a healthier food environment, but also through health promotion efforts that seek to improve population diets. The World Health Organization (WHO) has recommended limiting the levels of nutrients of concern in products to ensure that consumers can access and afford healthy food choices through manufacturers’ product portfolios [6].

Nutrient profiling is defined by the WHO as “the science of classifying or ranking foods according to their nutritional composition for reasons related to preventing disease and promoting health” [7]. Nutrient profiling models use algorithms that take into consideration the amounts or the presence of nutrients and other components of a food product to characterize its degree of “healthfulness” through either numerical scores or qualitative classifications (e.g., eligibility to display health claims or, in some countries, requirements to add warning labels). Nutrient profiling is a tool to classify individual foods, not diets, but nutrient profile models are commonly used to underpin policies designed to improve the overall nutritional quality of the food supply, and is recognized by the WHO as a helpful method to use in conjunction with interventions aimed at improving the overall nutritional quality of diets [7]. Because most calories consumed by Americans derive from processed food and beverage sources, and because high levels of processing are associated with nutritionally-poor products [8,9], the level of processing among foods and beverages can also be included as part of an overall assessment of healthfulness. Classification of foods and beverages by degree of processing can provide insight into dietary factors that contribute to obesity risk and weight gain, as well as adverse health outcomes such as cardiovascular disease and all-cause mortality [10,11,12,13].

The objective of this study was to perform a cross-sectional assessment of the state of the US packaged food and beverage supply by reporting the nutritional composition and indicators of healthfulness and processing across the country’s largest food and beverage manufacturers. By doing so, this study sought to provide useful information for consumers, researchers, and policymakers to encourage food manufacturers to reformulate or replace unhealthy products and to inform the US government on where action may be needed to improve the healthfulness of the US packaged food and beverage supply.

## 2. Materials and Methods

### 2.1. Nutritional Data

Data for this study are from the largest publicly-available US branded food composition database. Label Insight’s Open Data initiative provides researchers with open access to granular food composition data. Researchers are granted the freedom to publish their findings based on Label Insight’s data without restriction. The database is updated daily and contains information on more than 300,000 barcoded food and beverage items, which represents >80% of all food and beverage products sold in the US over the past three years [14]. Nutritional product data were extracted from Label Insight’s database in January 2019. Data included the Universal Product Code (UPC), brand name, product description, energy content (calories/100 g), total fat (g/100 g), saturated fat (g/100 g), total sugar (g/100 g), sodium (mg/100 g), and “not a significant source of” statements. Where implausible nutrient values (defined as >100 g of total fat, saturated fat, or sugars per 100 g) were present, these values were redefined as missing. FDA guidelines allow nutrition facts panels to include “not a significant source of” statements if specific nutrient values fall below a threshold for labelling, disclosure, or both. These statements were used to redefine missing nutritional data (e.g., energy, dietary fiber, protein, saturated fat, sodium, and total sugar) as 0. Food and beverage products were categorized based on the system developed by the Global Food Monitoring Group [15]. This hierarchical system classifies foods into major categories (e.g., bread and bakery products) and minor categories (e.g., bread; biscuits).

### 2.2. Measures of Healthfulness and Level of Processing

There is no international consensus on the superiority of one particular nutrient profiling model, in part due to the different purposes and contexts for which each model has been developed. One of the most widely used nutrient profile models is that underpinning the Health Star Rating (HSR) system [16,17,18,19,20]. This system is a voluntary, interpretive, front-of-pack nutrition labelling system in place for packaged foods in Australia and New Zealand. The system is designed to assist consumers in making healthier choices and to stimulate the food industry to reformulate their product lines to make them healthier [21]. The underlying nutrient profile model assesses adverse nutrients (as the densities of energy, sodium, total sugars, and saturated fat) and positive nutrients (as the densities of fruit and vegetable content, protein, and fiber). The system scores products based on their nutritional composition per 100 g or 100 mL. These scores are then converted to a ‘Health Star Rating’ from ½ (least healthy) to 5 (most healthy) stars in ½ star increments [22].

Calculation of the HSR was performed using criteria specified in the “Guide to the Health Star Rating Calculator”, endorsed by the Australian government [22]. Each product was assigned to one of six HSR categories (Category 1: Beverages other than dairy beverages; Category 1D: Dairy beverages; Category 2: All foods other than those included in Category 1, 1D, 2D, 3 or 3D; Category 2D: Dairy foods other than those included in Category 1D or 3D; Category 3: Oils and spreads; Category 3D: Cheese and packaged cheese). The HSR was calculated by (1) assigning baseline points for energy, saturated fat, total sugars, and sodium content per 100 g; (2) awarding modifying points for fruit, vegetable, nut and legume (FVNL) content, protein and fiber where applicable; (3) calculating an overall score by subtracting modifying points from baseline points, with a lower score reflecting a more nutritious food product; and (4) assigning an HSR (from 0.5 to 5.0 stars in half-star increments) according to the overall score using the defined scoring matrix. Appendix A outlines an example of an HSR calculation. Products were identified as “healthy” if the HSR value was greater than or equal to 3.5 out of 5.0 stars [18].

The NOVA classification framework was used to assign a level of processing to each food and beverage item. The NOVA framework classifies food and beverage products into four groups according to the nature, extent, and purpose of industrial processing used in their production, and is the most widely used system for studying food and beverage processing [23]. The four groups include ‘unprocessed or minimally processed foods’ (e.g., fresh fruits and vegetables), ‘processed culinary ingredients’ (e.g., salt, sugar, and honey), ‘processed foods’ (e.g., canned vegetables, cheese, and salted nuts) and ‘ultra-processed food and drink products’ (e.g., carbonated drinks, breakfast cereals, cakes, and pastries). Products were mapped to one of the four NOVA Level of Processing codes by minor food category and were further dichotomized to “ultra-processed” or not [4]. The NOVA system has been demonstrated to have moderate agreement with the International Food Information Council and University of North Carolina at Chapel Hill classification systems. Within the NOVA system, sodium and added sugar content are predictive of categorical classification in multinomial regression [24].

### 2.3. Manufacturers

Market shares for food and beverage manufacturers in 2018 were identified using data reported by Euromonitor [25]. The separate reports for 2018 US packaged food products and soft drinks manufacturers were retrieved in December 2018. Euromonitor reports include a percentage for all companies with greater than 0.1% US market share by retail sales in any of the previous 10 years. All other companies are aggregated by Euromonitor into either “private label”, “artisanal”, or “others” as applicable. By Euromonitor reporting, there were 126 packaged food manufacturers and 46 soft drink manufacturers with greater than 0.1% US market share between 2009–2018. The reports were merged by manufacturer, and the US retail sales volumes of packaged foods ($395.3 USD billion) and soft drinks ($110.1 USD billion) in 2018 were used to calculate the combined 2018 US market share of each manufacturer. Each included manufacturer’s US website was searched for a list of brands owned, and products under each of those brands were identified in the main dataset. Manufacturer-specific analysis was restricted to the top 25 manufacturers in 2018, based on combined market share of packaged foods and soft drinks after the exclusion of private labels, manufacturers of primarily nutritional products, and manufacturers with fewer than 100 products in the dataset.

### 2.4. Exclusion Criteria

Products ineligible for analyses included: products with duplicate UPCs, products categorized as non-food and products that did not display a nutrition facts panel (NFP), and products that differed only by package size. Products missing nutrient information (e.g., energy, saturated fat, sugar, sodium, protein, and fiber) required to calculate the HSR were also excluded.

Non-food products included categories such as alcoholic beverages, herbs and spices, plain teas and coffees, and vitamins and supplements, because they were not eligible to display an HSR according to the HSR guidelines. Products that did not display an NFP and products missing nutrient information included: delicatessen-type food, fresh bakery products, and confectionery that are sold over the counter and in most cases prepared by the retailer; self-service bulk foods; fresh produce and seafood served over the counter or via the deli; packaged single-ingredient fish or game meat and some custom-processed fish and game; certain egg cartons; foods manufactured by small business; and foods with very small packages (confectionery), among others.

### 2.5. Analysis

Graphical representations of the data were inspected to assess the distributions of HSRs and levels of nutrients. The mean HSR was calculated overall for each major and minor food category in the entire dataset and for each manufacturer. The proportions of products considered “ultra-processed” and receiving an HSR of 3.5 or greater were determined overall, and by category and manufacturer. The HSR value of 3.5 was chosen as an indicator of healthfulness, as this cut-off has been used in numerous previous studies [19,20,26]. Because nutrient data were not normally distributed, the median (Interquartile Range (IQR)) for energy density (kcal/100 g), saturated fat (g/100 g), total sugars (g/100 g), and sodium (mg/100 g) were calculated as secondary outcomes for each major and minor food category in the dataset overall and for each manufacturer. For visual comparison between categories, deciles for nutrient values among all products were calculated individually for energy density, saturated fat, sodium, and total sugar. Deciles were mapped to a color palette and the median nutrient value of each category. For statistical analyses, SAS version 9.4 (SAS, Cary, NC, USA) and R version 3.5.1 (R Foundation, Vienna, Austria) were used.

## 3. Results

After removal of ineligible products, including products with duplicate UPCs (*n* = 77), products categorized as non-food and products that did not display an NFP (*n* = 31,679), products that differed only by package size (*n* = 17,728), and 47,704 (17%) products missing nutrient information required to calculate the HSR, the final sample included 230,156 food and beverage products (Appendix A). The number of products in each major category ranged from 487 in Eggs to 38,032 in Fruit, vegetables, nuts and legumes (Table 1).

### 3.1. Health Star Rating and Level of Processing

The mean HSR for all US packaged food and beverage products was 2.7 (standard deviation (SD = 1.4) from a possible maximum of 5.0 stars (Table 1). HSRs varied by major food category with *Eggs* having the highest mean HSR (3.9 (SD = 0.5)) followed by Fruit, vegetables, nuts and legumes (3.7 (SD = 1.1)) and Seafood and seafood products (3.7 (SD = 0.8)). Confectionery had the lowest HSR of all categories examined (1.1 (SD = 0.7)), followed by Sugars, honey, and related products (1.5 (SD = 0.9)). The distribution of HSRs varied across major categories (Figure 1). Bimodal and trimodal distributions observed in Dairy, Non-alcoholic beverages, and Meat and meat alternatives were driven by variability within minor food categories (Appendix A).

Eggs, Seafood and seafood products, and Fruit, vegetables, nuts and legumes had the highest proportion of products considered ‘healthy’, with an HSR of 3.5 or greater (94%, 82%, and 70% respectively). The overall mean proportion of products considered ultra-processed was 71%. When examining results by category, the level of processing did not always correlate with products considered healthy using the HSR. For example, although Eggs had the highest proportion of products considered healthy using the HSR (94%), and also had the lowest proportion of products considered ultra-processed under NOVA (0%), Edible oils had the next lowest proportion of ultra-processed foods (8%), yet only 56% of these products were considered healthy using the HSR. Not surprisingly, Fruit, vegetables, nuts and legumes had a low proportion of ultra-processed products (18%), and 100% of Snack foods were considered ultra-processed.

### 3.2. Nutrient Composition

Across all US packaged food and beverage products, the median energy content was 271 kcal/100 g (IQR 92–400), the median saturated fat content was 0.9 g/100 g (IQR 0.0–6.1), the median sodium content was 250 mg/100 g (IQR 31–536), and the median total sugars content was 6.2 g/100 g (IQR 1.4–24.6). (Table 2). Edible oils had the highest median energy content (800 kcal/100 g (IQR 714–857)) and saturated fat content (13.3 g/100 g (IQR 10.7–21.4)) out of all the major food and beverage categories. Meat and meat alternatives had the highest median sodium content (788 mg/100 g (IQR 509–1107)) and Sugars, honey, and related products the highest median total sugar content (66.7 g/100 g (IQR 45.0–82.9)). Bread and bakery products was the only category to consistently be among the highest 1/3 across all 4 nutrient categories in terms of nutrient levels (energy, saturated fat, total sugars, and sodium). Considerable variability was demonstrated between minor categories within Fruits, vegetables, nuts and legumes, Foods for specific dietary use, Dairy, Non-alcoholic beverages, Meat and meat alternatives, and Bread and bakery products (Appendix A). Sodium values were highly right-skewed across all major categories (Appendix A), therefore no median sodium value within a major category fell within the highest decile.

### 3.3. Results by Manufacturer

The top 25 eligible manufacturers included in these analyses accounted for 42.8% of total 2018 retail sales, but just 28,808 (12.5%) of all products within the dataset were used for analysis. The number of products in the dataset per manufacturer ranged from 100 for Manufacturer A to 3831 for Manufacturer E (Table 3). Compared with the overall food supply, food products produced by these manufacturers, which accounted for nearly half of US retail sales, had modestly lower mean HSR (2.5 (SD = 1.3) vs. 2.7 (SD = 1.4); *p*-value < 0.0001) and substantially higher frequency of ultra-processing (86% vs. 71%; *p*-value < 0.0001).

There was considerable heterogeneity in the healthfulness of food and beverage products among manufacturers. For example, Manufacturer A, which mainly sold dairy products, (Figure 2) had the highest mean HSR (4.6 (SD = 0.7)), and Manufacturer Y, which mainly sold confectionery products, had the lowest mean HSR (0.9 (SD = 0.7)). Manufacturer A also had the highest proportion of products with an HSR of 3.5 or greater (94%), nearly 30% more healthy products than the next ranked manufacturer (Manufacturer B; 66%). Only one manufacturer had <50% of products considered ultra-processed (Manufacturer A; 26%). Healthfulness by manufacturer is highly dependent on product portfolio and concentration within a specific minor category can be an important driver of relative healthfulness. Both Manufacturers D and V have a high proportion of bread and bakery products (94% and 69% respectively). However, differences in their product portfolios by minor category (Manufacturer D: 73% bread, 17% cakes, muffins, and pastries, and 4% crackers and cookies vs. Manufacturer V: <1% bread, 2% cakes, muffins, and pastries, and 67% crackers and cookies) are drivers of substantial differences in overall HSR (2.9 (SD = 1.2) vs. 1.7 (SD = 1.1), respectively). Across all products (Appendix A), the mean HSR of bread (3.3 (SD = 0.8)) is roughly twice the mean HSR of crackers and cookies (1.6 (SD = 1.0)).

## 4. Discussion

We found that the US packaged food and beverage supply is very large and generally unhealthy, with a high proportion of ultra-processed foods produced by the top 25 food and beverage manufacturers. Among the 230,156 packaged food and beverage products examined, the mean HSR was 2.7 out of 5.0, and varied substantially by major food category. Seventy-one percent of products were classified as ultra-processed using the NOVA classification, and 40% of products had an HSR ≥3.5. Our findings are in line with previous research from both the US and other western countries [4,27]. In comparison to Australia, for example, for which there exists a comparable published state of the food supply report [27], the US food supply has a similar mean (SD) HSR (2.7 (1.4) US vs. 2.8 (1.4) Australia), a similar proportion classified as ‘healthy’ (HSR ≥ 3.5; 40.2% US vs. 39.2% Australia), yet a higher proportion of highly processed products (70.9% ultra-processed in US vs. 60.5% highly processed Australia). US products also had a lower median saturated fat content (g/100 g; 0.9 US vs. 1.7 Australia), higher median total sugar content (g/100 g; 6.2 US vs. 5.3 Australia), and higher sodium content (mg/100 g; 250 US vs. 163 Australia). A recent global report by the Access to Nutrition Foundation, focusing on the top 22 global food and beverage manufacturers, found that the US had one of the highest mean HSRs of all countries included (2.6), with middle income countries such as India and China much lower (2.1 and 1.8 respectively), which demonstrates the general unhealthiness of the global food supply [28].

Out of the top 25 manufacturers included in this analysis, only one manufacturer had <50% of products considered ultra-processed, and 14 manufacturers had >90% of products considered ultra-processed. Manufacturers that had their portfolios dominated by dairy products generally fared better (e.g., Manufacturer A), and those with portfolios dominated by confectionery items scored poorly (e.g., Manufacturer Y). This is in line with a recent report by the Access to Nutrition Foundation which ranked the top 22 global food and beverage manufacturers, which also found that global dairy companies generally performed better in terms of the healthiness of product portfolios than other manufacturers [28]. Recent research from the US examining the contribution of processed and convenience food categories to purchases by US households found that almost two thirds of energy content in purchases by US households came from ultra-processed food sources, the majority of which food items exceed recommended limits for saturated fat, sugar, and sodium content [4]. Ultra-processed foods dominate food supplies in a large number of western countries, and increasingly in low- and middle-income countries [29]. This, along with the current findings, is cause for concern, with US obesity levels and rates of chronic disease still heavily burdening the US health system. Research has shown that a significant positive association exists between national household availability of ultra-processed foods and national prevalence of obesity among adults [30].

The top 25 manufacturers in the US accounted for less than half (42.8%) of retail sales and only 28,808 (12.5%) of products. The US food and beverage market is both highly heterogeneous by product category and highly fragmented with smaller manufacturers accounting for a large proportion of products measured by sales volume. Despite the breadth of manufacturers, the relatively high concentration of sales among top manufacturers reflects a potential opportunity for widespread effect, if these manufacturers could be influenced to improve the healthfulness of a relatively small number of commonly purchased products. Furthermore, many food and beverage manufacturers operate in a number of different countries, and there is considerable overlap among manufacturers who sell products globally, so potential exists for healthier products to be manufactured and sold even within a single manufacturer or brand. For example, products in the US have been shown in previous studies to have higher levels in sodium compared to the same products sold in a different country [31,32]. Longitudinal surveillance of the global food and beverage supply through a standardized platform would provide an ongoing assessment of the foods and beverages that are available and sold, as well as their distribution. A monitoring system such as this could help policymakers more clearly understand what is in the global, regional, and national food and beverage supplies, which has potential implications in influencing large-scale policy action to curb the epidemic of diet-related ill health.

This report has several strengths, including being the largest, most contemporary report of its kind. Our analysis included data from the largest publicly-available US branded food composition database, representing over 80% of all packaged food and beverages in the US. Because the US has the largest food and beverage supply in the world, this report adds critical research on the healthfulness of the global food and beverage supply. This report also has limitations, including not capturing 100% of the products available in the US food and beverage supply, particularly fresh and unpackaged food products. The findings here are likely relevant to the majority of products that are consumed, the retail sales for packaged food and beverage products in 2018 captured within our manufacturer-level data was $505.4 USD billion. This report also excluded 17% of products in the Open Data Initiative database due to missing data, however, it seems likely the overall results would be similar, given the potential explanations for missing data. Determination of the cause for the missing data was outside the scope of this analysis, but merits further investigation to determine if these products were exempted from listing an NFP (e.g., delicatessen-type food, bakery products, and confection that are sold over the counter and in most cases prepared by the retailer; self-service bulk foods; fresh produce and seafood served over the counter or via the deli; packaged single-ingredient fish or game meat and some custom processed fish and game; certain egg cartons; foods manufactured by small business; foods with very small packages) or missing single or multiple nutrition values due to omissions or labelling errors. The use of the HSR system might also be considered a potential limitation because it is not currently used in the US, however, the HSR is a widely used and accepted nutrient profiling system which allows comparison across product categories, and we complemented the use of HSR with analyses using the NOVA classification framework.

## 5. Conclusions

The US packaged food and beverage supply is large, highly processed, generally unhealthy, and heterogeneous. The wide variation in healthfulness and levels of processing observed within and across food and beverage categories demonstrates that there exist opportunities for large-scale improvements in the US food supply. Data collection of packaged food and beverage products is challenging due to constant evolution and replacement of products and the sheer size of the US packaged food and beverage supply. Complete and current coverage requires improved and continued surveillance including use of novel sources such as crowd-sourcing to capture real-time information on the food supply.

## Figures and Tables

**Figure 1 nutrients-11-01704-f001:**
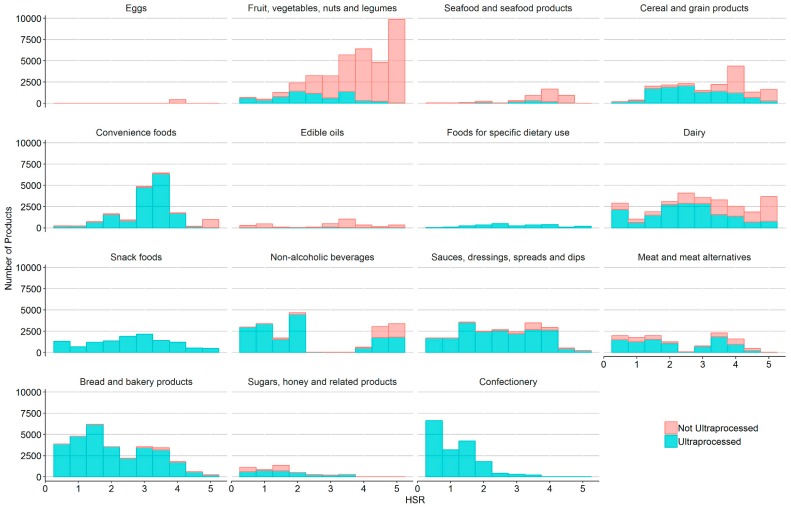
Distribution of Health Star Ratings (HSRs) by major food category for foods that were (blue) and were not (pink) ultra-processed.

**Figure 2 nutrients-11-01704-f002:**
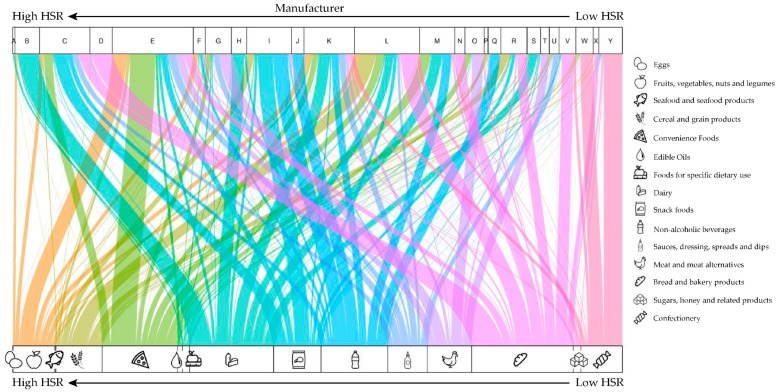
Alluvial plot demonstrating top 25 (A–Z) US manufacturers’ product portfolio compositions by mean Health Star Rating (HSR) and major food category. Colors represent major food categories identified by icons.

**Table 1 nutrients-11-01704-t001:** Healthfulness of the US food and beverage supply by major food category.

Major Food Category	No. Products	Nutrient Profiling Summary Score		Level of Processing
		Mean HSR (SD)	Proportion ′Healthy′ HSR ≥ 3.5 (%)	Proportion Ultra-Processed (%)
Eggs	487	3.9 ± 0.5	94.3	0.0
Fruit, vegetables, nuts and legumes	38,032	3.7 ± 1.1	70.3	18.3
Seafood and seafood products	4325	3.7 ± 0.8	81.7	20.3
Cereal and grain products	18,024	3.2 ± 1.1	52.8	61.8
Convenience foods	17,980	3.2 ± 0.8	51.9	90.9
Edible oils	3246	3.0 ± 1.3	55.9	8.4
Foods for specific dietary use	2369	2.9 ± 1.1	41.4	100.0
Dairy	27,839	2.9 ± 1.4	40.7	61.5
Snack foods	12,231	2.6 ± 1.2	29.6	100.0
Non-alcoholic beverages	19,954	2.5 ± 1.7	35.7	83.1
Sauces, dressings, spreads, and dips	21,772	2.5 ± 1.2	32.9	92.3
Meat and meat alternatives	12,249	2.2 ± 1.3	35.7	75.3
Bread and bakery products	30,194	2.1 ± 1.1	20.0	98.6
Sugars, honey, and related products	4625	1.5 ± 0.9	6.7	73.0
Confectionery	16,829	1.1 ± 0.7	1.4	100.0
All	230,156	2.7 ± 1.4	40.2	70.9

HSR, Health Star Rating. SD, standard deviation

**Table 2 nutrients-11-01704-t002:** Deciles of energy, saturated fat, sodium, and total sugars by major category in the US food and beverage supply.

Major Food Category	Nutrient Composition Colored by Decile (Median Interquartile Range (IQR))				
	Energy (kcal/100 g)	Saturated Fat (g/100 g)	Sodium (mg/100 g)	Total Sugars (g/100 g)	Nutrient Decile ^1^
Eggs	140 (125–140)	3.0 (2.0–3.3)	140 (140–163)	0.0 (0.0–0.0)	0
Fruit, vegetables, nuts and legumes	107 (48–421)	0.0 (0.0–3.3)	63 (4–300)	5.6 (2.4–25.0)	1
Seafood and seafood products	120 (89–189)	0.5 (0.0–1.8)	375 (223–529)	0.0 (0.0–0.9)	2
Cereal and grain products	368 (353–394)	0.0 (0.0–2.3)	221 (0–500)	4.4 (1.8–23.1)	3
Convenience foods	176 (103–247)	1.7 (0.4–3.5)	384 (265–557)	2.1 (1.1–3.6)	4
Edible oils	800 (714–857)	13.3 (10.7–21.4)	0 (0–0)	0.0 (0.0–0.0)	5
Foods for specific dietary use	380 (216–417)	4.0 (0.5–6.7)	240 (84–380)	15.0 (5.1–26.9)	6
Dairy	206 (88–321)	6.0 (0.7–13.3)	88 (50–563)	7.5 (0.0–19.4)	7
Snack foods	500 (438–536)	3.6 (1.8–7.1)	607 (411–893)	3.6 (0.0–7.1)	8
Non-alcoholic beverages	40 (12–51)	0.0 (0.0–0.0)	8 (0–17)	8.3 (0.4–11.8)	9
Sauces, dressings, spreads and dips	125 (50–264)	0.0 (0.0–2.9)	567 (342–967)	5.6 (1.7–17.2)	
Meat and meat alternatives	232 (143–307)	4.2 (0.9–8.9)	788 (509–1107)	1.1 (0.0–2.9)	
Bread and bakery products	386 (284–447)	3.6 (0.0–7.9)	389 (267–526)	20.0 (4.7–34.3)	
Sugars, honey and related products	333 (267–375)	0.0 (0.0–0.0)	0 (0–56)	66.7 (45.0–82.9)	
Confectionery	400 (350–500)	2.5 (0.0–16.7)	60 (12–128)	54.8 (45.2–66.7)	
All	271 (92–400)	0.9 (0.0–6.1)	250 (31–536)	6.2 (1.4–24.6)	

^1^. Deciles are calculated individually by nutrient among all 230,156 food and beverage products. Color for each major category and nutrient are applied based on the decile of the median.

**Table 3 nutrients-11-01704-t003:** Healthfulness of the US food and beverage supply among the top 25 manufacturers, based on 2018 retail sales.

ID	Nutrient Profiling Summary Score		Level of Processing	2018 US Market Share, %	Products in Database	
	Mean HSR (SD)	Proportion ′Healthy′ HSR ≥ 3.5 (%)	Proportion Ultra-Processed (%)		No.	%
A	4.6 ± 0.7	94.0	26.0	0.39	100	0.04
B	3.6 ± 1.4	65.5	68.3	0.98	1165	0.51
C	2.9 ± 1.3	47.4	88.9	1.67	2388	1.04
D	2.9 ± 1.2	45.4	95.5	0.78	1048	0.46
E	2.9 ± 1.2	48.7	75.4	1.56	3831	1.66
F	2.8 ± 1.3	40.9	76.2	0.47	563	0.24
G	2.8 ± 1.2	52.1	86.3	0.70	1238	0.54
H	2.7 ± 1.4	32.6	64.7	0.55	726	0.32
I	2.7 ± 1.2	32.1	90.8	8.48	2109	0.92
J	2.4 ± 1.5	27.6	81.6	4.66	591	0.26
K	2.4 ± 1.4	32.1	72.9	3.92	2385	1.04
L	2.4 ± 1.2	26.7	88.0	1.80	3082	1.34
M	2.3 ± 1.3	27.1	94.0	3.90	1658	0.72
N	2.3 ± 1.3	28.5	97.3	0.39	484	0.21
O	2.3 ± 1.1	33.1	94.5	1.09	910	0.40
P	2.2 ± 0.8	6.1	99.4	0.39	180	0.08
Q	2.1 ± 1.5	23.0	91.4	1.70	608	0.26
R	2.1 ± 1.0	15.0	99.8	1.80	1236	0.54
S	2.0 ± 0.8	6.7	96.4	1.11	638	0.28
T	1.9 ± 1.3	24.9	95.4	0.39	410	0.18
U	1.9 ± 1.1	15.5	89.1	0.39	466	0.20
V	1.7 ± 1.1	12.1	99.1	1.64	791	0.34
W	1.2 ± 1.0	9.3	98.7	1.80	820	0.36
X	1.1 ± 0.7	1.5	100.0	0.39	263	0.11
Y	0.9 ± 0.7	1.8	99.2	1.88	1118	0.49

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
