# Peer review of "The Healthfulness of the US Packaged Food and Beverage Supply: A Cross-Sectional Study"

_nutrients, 2019, doi:10.3390/nu11081704_

Reviewer 1 Report

This is a very well written and presented study.

Please provide references for the following sentences:

Line 38 – 41: The World Health Organization (WHO) has recommended limiting the levels of nutrients of concern in products to ensure that consumers can access and afford healthy food choices through manufacturers’ product portfolios.

Line 248: Our findings are in line with previous research from both the USA and other western countries.

Typo in Table 3: :2108 US Market Share” Should this be 2018?

Author Response

REVIEWER COMMENTS: 

Reviewer #1 

1. This is a very well written and presented study.

Response: Thank you.

Changes to manuscript: None.

2. Please provide references for the following sentences:

Line 38 – 41: The World Health Organization (WHO) has recommended limiting the levels of nutrients of concern in products to ensure that consumers can access and afford healthy food choices through manufacturers’ product portfolios.

Line 248: Our findings are in line with previous research from both the USA and other western countries.

Response: We now include references for these two lines.

Changes to manuscript (References):

1.     Add new reference “Diet, nutrition and the prevention of chronic diseases. World Health Organ Tech Rep Ser 2003, 916, i-viii, 1-149, backcover” for Lines 38-41

2.     Add existing references “Poti, J.M.; Mendez, M.A.; Ng, S.W.; Popkin, B.M. Is the degree of food processing and convenience linked with the nutritional quality of foods purchased by US households? Am J Clin Nutr 2015, 101, 1251-1262, doi:10.3945/ajcn.114.100925” and “Crino, M.; Sacks, G.; Dunford, E.; Trieu, K.; Webster, J.; Vandevijvere, S.; Swinburn, B.; Wu, J.Y.; Neal, B. Measuring the Healthiness of the Packaged Food Supply in Australia. Nutrients 2018, 10, doi:10.3390/nu10060702.” to line 248

3. Typo in Table 3: “2108 US Market Share” Should this be 2018?

Response: We have corrected this typo.

Changes to manuscript (Table 3): Change “2108” to “2018”

Reviewer 2 Report

An interesting paper, using readily available data. 

There is a spelling mistake on page 8, line 230.

Otherwise, I only have comments to make. 

Healthfulness is subjective - agreed there are levels of sodium, sugar, saturated fat etc, however, the use of artificial sweeteners could be seen as a health risk and it depends on how much people eat the processed foods per day. In addition, putting all meat and meat products together would skew results - fresh meat has little processing, whereas ham and meat pies are highly processed, but could still be healthy if eaten with lots of fruit and vegetables. Probably not in the remit of the paper but a sentence on how much sales of the unhealthy foods compared to healthy food might add context. 

Comparing manufacturers with like-for-like commodities, such as bread, that can be very healthy or unhealthy might be useful, but comparing to manufacturers of confectionary, very unhealthy, is nonsensical. 

Author Response

Reviewer #2 

1. An interesting paper, using readily available data.

Response: Thank you.

Changes to manuscript: None.

2. There is a spelling mistake on page 8, line 230.

Response: We have corrected this typo.

Changes to manuscript (Results by Manufacturer): Change “Heathfuless” to “Healthfulness”.

3. Healthfulness is subjective - agreed there are levels of sodium, sugar, saturated fat etc, however, the use of artificial sweeteners could be seen as a health risk and it depends on how much people eat the processed foods per day. In addition, putting all meat and meat products together would skew results - fresh meat has little processing, whereas ham and meat pies are highly processed, but could still be healthy if eaten with lots of fruit and vegetables. Probably not in the remit of the paper but a sentence on how much sales of the unhealthy foods compared to healthy food might add context.

Response: Our aim in this analysis was to present the state of the food supply, including food and beverage products available for purchase by consumers. We agree that healthfulness is a nuanced, complex, and evolving concept and that the presence of unhealthy or healthy products in the food supply is not necessarily indicative of consumption. However, we use the Health Star Rating classification system, Australia’s front-of-pack labeling nutrient profiling system, as a framework for healthiness and complement this with an estimate of level of processing using the NOVA framework to capture related, but at times distinct, domains to which the reviewer refers. We chose to focus on the top 25 manufacturers by sales volume to illuminate products which are more likely to be commonly purchased. The sales data to which we had access was not granular enough to include information on the sales of healthy versus unhealthy food. We include Table 3 and Figure 2 to help identify the relative healthfulness of each manufacturers’ product portfolio. Within the discussion, we have modified wording of the limitations to underline the scope of the analysis and exclusion of fresh and unpackaged food products, as well as the sales volume (also stated in the methods; line 132) of the products that were included within the analysis.

Changes to manuscript: Change “This report also has limitations, including not capturing 100% of the products available in the US food and beverage supply, though the findings here are likely relevant to the majority of products that are consumed.”

To: “This report also has limitations, including not capturing 100% of the products available in the US food and beverage supply, particularly fresh and unpackaged food products. The findings here are likely relevant to the majority of products that are consumed, the retail sales for packaged food and beverage products in 2018 captured within our manufacturer-level data was $505.4 USD billion.”

4. Comparing manufacturers with like-for-like commodities, such as bread, that can be very healthy or unhealthy might be useful, but comparing to manufacturers of confectionary, very unhealthy, is nonsensical.

Response: We chose to focus our analysis on a comprehensive overview of the food supply and thus comparisons across manufacturers of varied food categories were part of the paper’s objective.  We disagree that such presentation of results, which inherently lead to comparisons, are “nonsensical” as the reviewer suggests. We agree with the reviewer that targeted comparisons of manufacturers within categories are useful, and members of our author team have reported on deeper analysis of specific food products, such as bread (Coyne K, et al. Pub Health Nutr. 2018; 21(3):632-636). Our results and discussion focus on the heterogeneity in food and beverage products among and within the top 25 US manufacturers in 2018. We include within our description of results that “…healthfulness by manufacturer is highly dependent on product portfolio.”

Changes to manuscript: None